# Insights into the Impact of Organizational Factors and Burnout on the Employees of a For-Profit Psychiatric Hospital during the Third Wave of the COVID-19 Pandemic

**DOI:** 10.3390/ijerph21040484

**Published:** 2024-04-15

**Authors:** Michael Seyffert, Chunyi Wu, Gülru F. Özkan-Seely

**Affiliations:** 1School of Business, University of Washington Bothell, Bothell, WA 98195, USA; gulru@uw.edu; 2Michigan Medicine, University of Michigan, Ann Arbor, MI 48109, USA; wuchunyi@umich.edu

**Keywords:** mental health, burnout, commitment, leadership, job satisfaction, management

## Abstract

In this paper, we provide insights into the interplay among the organizational, job, and attitudinal factors and employees’ intentions to resign during the third wave of the COVID-19 pandemic at a mental health hospital. We point out shortcomings in the relationship dynamics between executive administration and operational staff and propose a pathway to develop more effective leadership frameworks to increase job satisfaction. We integrate qualitative data from case information and open-ended questions posed to employees at a mental health hospital and quantitative data from a small-scale survey (*n* = 19). We highlight that the ability to achieve objectives, work autonomy, burnout, affective commitment, distributive and procedural justice, and job satisfaction are critical in determining individuals’ intentions to resign. Individuals identified disconnectedness and moral distress as critical aspects, while highlighting empathy, compassion, satisfaction, and confidence as pivotal elements. Mental healthcare settings could benefit from enhancing the staff’s ability to achieve objectives, work autonomy, affective commitment, and both distributive and procedural justice. Addressing burnout and implementing measures to increase job satisfaction are equally vital. Efficiently restructuring dynamics between various leadership levels and staff can significantly improve employee retention.

## 1. Introduction and Background

Job-related burnout (referred to as burnout hereafter), defined as a state of emotional, physical, and mental exhaustion caused by prolonged work-related stress, has long been a critical area of concern for hospitals. Its prevalence among healthcare professionals in the US is well-reported, with nurses experiencing burnout rates ranging from 35.0% to 45.0% and physicians, approximately 50.0% [1,2]. There are significant challenges associated with burnout, such as emotional exhaustion, job satisfaction, work safety, psychological empowerment, and the meaningfulness of work, that plague healthcare professionals [3,4,5,6]. In their qualitative study using triangulation methods, including interviews with focus groups composed of supervisors and trainees, [7] indicate that “participants describe burnout as an insidious syndrome lying on a spectrum, with descriptions coalescing under seven themes: altered emotion, compromised performance, disengagement, dissatisfaction, exhaustion, overexertion, and feeling overwhelmed.” Historically, periods of increased burnout have been associated with an increase in medical errors and poorer patient outcomes [8,9,10,11].

In addition to burnout, personnel operating in psychiatric healthcare environments are prone to emotional exhaustion, which is the depletion of their emotionality and the loss of their enthusiasm over their workplace duties [12], and to compassion fatigue (CF), which is the stress and negative coping behaviors resulting from caring for traumatized individuals. These can lead to emotional and physical distress [13], including moral distress, where “one knows the right thing to do, but institutional constraints make it impossible to pursue the right course of action” [14,15], a factor especially prevalent among psychiatric nurses [16,17,18]. A key factor that differentiates burnout from these phenomena is that burnout represents sustained frustration or failure in achieving goals. One way that emotional and compassion satisfaction can be managed is through perceiving stress as a manageable component of self-care [19].

Burnout and CF are concepts that overlap to an extent where each only partially explains the physical and mental exhaustion typically related to job stress. Job characteristics such as moderated workloads [20] and availability of resources [21] combined with the personal characteristics of employees such as their degree of their mindfulness [22], psychological hardiness [23], and rumination [24] create unique environments that drive the degree that these factors impact job stress and engagement. When combined, burnout and CF can result in job absenteeism and diminished job engagement, impaired decision-making, and ultimately, reduced job satisfaction [25], especially in psychiatric healthcare settings [26,27]. Job engagement, whose drivers include factors shaped by organizational structures and features such as trust, professional attitudes, and psychological characteristics, is a key determinant in the level of job satisfaction of an employee [28,29,30]. Job satisfaction is a critical indicator when assessing the likelihood of employees leaving their job, as evidenced by employees’ low job satisfaction ratings being associated with their intention to turnover [31,32,33,34]. Job satisfaction and employee turnover are key metrics for HR management in healthcare. As a result, factors affecting the metrics such as burnout and CF are focal concerns for leadership, as is the need to improve work conditions to avoid staffing shortages and foster a stable and positive interdisciplinary workforce [35,36].

The arrival of the COVID-19 pandemic represented a global public mental health crisis. Rates of anxiety, depression, and isolation increased [37,38]. Research identified several knowledge and behavioral factors as key mediators contributing to an individual’s sense of well-being. For example, individuals with more direct experience of the virus were found to have an increased “sense of threat,” which correlated with increased psychological distress and decreased well-being. Yet, applying knowledge of COVID-19 to engage in protective behaviors was found to decrease psychological distress and have a positive effect on well-being [39].

Shortly after the onset of the COVID-19 global pandemic, there was a recognition of a “parallel mental health crisis” facing healthcare workers (HCWs). Across the nation, the mental stress of HCWs, particularly those in acute settings such as hospitals, intensified significantly [40]. The heightened risk of physical illness due to COVID-19 infection, coupled with the potential for the virus to spread beyond the workplace, further exacerbated employees’ frustration, isolation, safety concerns, and chronic job-related stress [41,42,43,44,45]. This, in turn, has been fueled by increased workplace obligations, which have contributed to declines in staff health, employee engagement, and organizational commitment [46], and escalated turnover of employees [47]. The pandemic has exerted a disproportionately severe toll on the global mental health workforce, leading to a cascade of negative consequences including heightened workloads, insufficient staffing levels, profound moral distress, and widespread burnout [48,49,50,51,52,53].

While the arrival of COVID-19 exacerbated the negative impact of work-related factors on caregiving staff, overwhelmed healthcare organizations, and sparked a surge in staff departures [54], it also crystallized the burden of mental health and the stress associated with the physical and emotional demands on HCWs that is an inherent part of such environments even in non-pandemic times. A review of 32 articles by [55] revealed that “long working hours, loneliness, inadequate rest and self-care, feelings of helplessness, pain of losing multiple patients every day, facing violence, depression, post-traumatic stress disorder” have plagued “frontline” HCWs including nurses, allied healthcare workers, and medical residents and physicians.

For-profit psychiatric organizations have historically faced financial challenges and have placed significant demands on their employees, particularly in terms of organizational citizenship behaviors (OCBs)—responsibilities that bolster a hospital’s resiliency [56] and its crisis response effectiveness [57]. These demands have led employees to call for greater support and consideration from organizational management [57]. As the COVID-19 pandemic advanced, the elevated risk of physical illness, intensified workplace responsibilities, and mounting stress led employees to feel a profound sense of neglect [58,59]. More recently, despite their increase in market share over the years, for-profit psychiatric hospitals have faced scrutiny due to unsafe conditions and practices, poor patient safety, financial improprieties, and closures [60], factors that amplify the psychological job-related distress of HCWs in these environments. As a result, it became incumbent upon ethically minded organizational leaders to find ways of improving social support [61] and reducing burnout through ‘‘health promoting leadership” [62], thereby improving employee self-efficacy and employee engagement [63].

In summary, burnout, job disengagement, compassion fatigue, and job-related stress significantly erode employees’ satisfaction, well-being, and their intention to remain in their positions [64], and these effects have been amplified by the spread of COVID-19. Leaders are compelled to reevaluate the extent to which the healthcare system is overwhelmed, both at the “bottom-up” frontline staff level and at the “top-down” administrative staff and executive leadership level, and to recognize the urgent need for a more positive and humane work culture. Such a culture should be grounded in principles of ethical and “sustainable leadership” [65] that is firmly committed to a set of values prioritizing communication, compassion, and interpersonal connections, rather than being dominated by a high-pressure environment that is exclusively performance-driven [66]. Leaders must grasp that fostering compassionate understanding and proactively shaping the ways in which people collaborate is essential. This approach requires more than simply applying knowledge; it demands a fundamental shift in leadership mentality and practice [67].

While the importance of understanding the factors that drive the burnout and compassion fatigue, job-related stress, and job engagement and satisfaction are well-accepted, there remains a gap in the literature that brings together these constructs and applies it to HCWs in mental healthcare settings through the utilization of mixed methods. We aim to fill this gap. In our study, we combine qualitative data in the form of case information and materials on personal experiences such as narrative statements about employees’ lived experiences and quantitative survey data [68]. We share viewpoints into the relationships between three critical factors associated with organizational factors (work autonomy, ability to achieve objectives, distributive and procedural justice attitudinal) and psychological (burnout) and attitudinal (job satisfaction) factors associated with the employees. In addition, we provide insights into the employees’ thought processes concerning their levels of work engagement and personal commitment. As such, we aim to shed light into relationships between these factors and the likelihood that employees would prefer to remain employed in their current organization or seek employment elsewhere, which we refer to as their intention to turnover.

## 2. Case: A CMO’s Assessment during Crisis of Compassion and Leadership

This study was motivated by the challenges faced in the psychiatric mental health environments, specifically those observed at the Voice Behavioral Hospital (VBH) (names of people and places have been changed to preserve anonymity in this healthcare setting). Since it opened its doors in spring 2019, VBH had been struggling. For the better part of a year, there existed a revolving door of chief nursing officers (CNOs), chief executive officers (CEOs), and chief medical officers (CMOs) that served only to increase organizational chaos and sap staff morale. Subsequently, during 2020, the world experienced the onset and surge of the COVID-19 pandemic. The CMO of a neighboring mental health hospital, the largest for-profit psychiatric hospital in the region and just a hundred miles or so north of VBH, was quite familiar with the numerous straining impacts associated with the COVID-19 pandemic on the healthcare workers. He had witnessed first-hand how problems such as nursing and provider shortages, absenteeism, workplace violence, and resignations could have a cumulative effect and erode both organizational and personal resources. This inflation of the responsibilities of the CMO, including those associated with leadership, was especially prevalent during the COVID-19 pandemic, and the situation at VBH had paralleled this trend.

Traditionally, CMOs operating in hospital settings have been tasked with ensuring that the highest-quality medical care is provided, problems associated with the safety of the staff and the patients are addressed in a timely manner, and the providers’ performances are supervised fairly and effectively. Over the years, the job description of CMOs has expanded, and the clinical and organizational activities this role is tasked with have surged. Therefore, “lack of both individual and organizational clarity regarding the roles, responsibilities and expectations of the position and lack of support for the position” have emerged as “major factors that limit success in the CMO position” [69,70] provides detailed information on the many hats worn by the CMOs and the staff in hospital settings and provides an illustrative example of the role CMOs play within the organizational structures of hospitals (Figure 1). Additional roles mentioned in Figure 1 are chief executive officer (COO), chief financial officer (CFO), chief medical informatics officer (CMIO), vice president of care coordination, chief quality officer (CQO), the chief of service (COS), and the medical executive committee (MEC). Length of stay (LOS), plan of care (POC), prior authorization (PA), and resource use are key utilization metrics tracked by the CMOs. These, along with the peer review credentialing, ongoing professional practice evaluation (OPPE), and careful monitoring and improvement of quality and safety, ensures effective management at the CMO level.

In 2021, during the third wave of the COVID-19 pandemic, the former CMO of the neighboring hospital joined VBH as their CMO. Already mindful of the potential confusion and conflict that may come with this role, he recognized his responsibilities at VBH were divided. In addition to his CMO duties, he oversaw managing a specialized psychiatric unit within VBH, which significantly intensified the expectations of his post, and, thus, severely restricted the time he had available for administrative duties. From an organizational perspective, as part of an executive leadership team, he was required to prioritize decisions based on the considerations at the hospital level as well as at the level of the healthcare provider system of which VBH was part (which we refer to as the “corporate” level). In addition, he had to adapt to any unforeseen demands on hospital resources, whether they were related to the day-to-day surges in unit acuity or the uptick in COVID-19 infections that might lead to changes in critical metrics associated with hospital utilization and quality of care, such as the status of available beds and changes in the patients’ length of stay (LOS). He understood that some barriers to effective organizational management were universal and could be applied to all psychiatric hospitals, while others were specific to the healthcare culture at VBH.

At VBH, the emotional cost of heightened health risks, neglect of self-care, and lack of compassion manifested in increased conflict between the inter-professional teams of mental health technicians, intake social workers, nursing, psychiatric nurse providers, and physicians. Providers were upset at intake staff, who were not adequately trained and lacked oversight, which led to a higher acuity of patients and decreased safety for patients and employees. Due to chronic understaffing, an external psychiatric provider telehealth team was hired by VBH corporate; however, nurses and providers were discouraged by the ineffectiveness of the telehealth systems in addressing the service gaps. Specifically, they noted that the externally contracted telehealth psychiatric nurse practitioners (PNPs) were stretched too thin, resulting in significant delays in patient appointments or, in some cases, an inability to see patients at all. Moreover, when the telehealth team did manage to see patients, albeit late, they were frequently perceived as lacking compassion and empathy. Frequently, the hospital medical records department experienced significant delays in receiving telehealth patient notes, which often arrived days or even weeks after the consultations. This not only increased the workload for providers, who were then responsible for seeing these neglected patients, but it also compromised the overall quality of patient care. There was also conflict among the hospital executives, including the CEO, CNO, and CMO, as well as the staff in the human resources department. Providers expressed frustration with the CEO regarding their salaries and escalating work demands, especially when they were tasked with seeing 20 or more patients per day. The CEO expressed concern to the CMO that he appeared to be more aligned with the providers than with the administration. In response, the CMO raised concerns with the CEO about inadequate training and safety measures in the patient intake process, as well as insufficient staffing and coverage on the units. In addition, the CMO recognized that the leadership and responsibility structure at VBH had clear shortcomings, exacerbated by the COVID-19 pandemic. The executive administration did not have a direct communication channel with the CMO, and, instead, information flow was through the CEO. The relationship of the CEO with the CMO was more of a top-down approach, which resulted in the scope and scale of responsibilities of the CMO being larger than that of an ideal setting. This ineffective structure is illustrated in Figure 2.

For the CMO, these conflicts served as clear indicators of an emerging leadership crisis. To proactively address this crisis, he recognized the need for adaptive leadership measures. He believed that the mission of the medical leadership at VBH should pivot to focus on the interrelated challenges of enhancing the quality of patient care, boosting staff engagement and satisfaction, and reducing intentions for staff turnover. To navigate these complexities, the CMO understood that a thorough assessment of work dynamics, staff, and organizational characteristics was essential, as these elements collectively contribute to staff well-being and a positive work environment. In addition, he would need to adopt a new leadership mindset, one that moved beyond technical solutions to a model of care that focuses on emotional solutions. Compassion, a foundational attitude in the establishment of a therapeutic alliance, is especially important in behavioral healthcare, as patients seeking psychological or psychiatric care are often not able to show themselves self-compassion and kindness. Additionally, the CMO viewed this crisis as an opportunity to collaborate with fellow executives, including the CNO, CEO, and members of the HR department. He aimed to leverage this collaboration to reprioritize and advocate for the recognition and autonomy of the staff. The CMO’s vision included expanding training opportunities or programs that acknowledge the “people capacity” within the organization. He believed that such initiatives could incentivize organizational commitment and promote a culture of compassion and resilience in the workplace.

Meanwhile, by summer 2021, top executives at the corporate level, and by extension, the hospital CEO, intensified their approach of cutting corners. They maintained a single-minded focus on increasing bed capacity and expanding programming, despite the glaring lack of adequately trained mental health staff to support these initiatives. This transactional approach to leadership not only undermined unit and staff safety but also eroded staff autonomy and compromised the sense of justice in the workplace. It created an environment where numbers and output were prioritized over the holistic well-being of patients and the job satisfaction and safety of the healthcare professionals entrusted with their care. Over the subsequent months, these manifestations of organizational and interpersonal dysfunction escalated, setting off a vicious cycle. This cycle was characterized by increasing emotional exhaustion among employees, widespread disengagement, and escalating staff turnover, further destabilizing the work environment and compromising patient care. Corporate directives came down aggressively, mandating that positions had to be cut, adding more tension to an already strained environment. Furthermore, despite the glaring inadequacy of staffing, expectations were set that the hospital should not only maintain its current services but also expand them. This included the ambitious plan to open an entire partial hospitalization program, which seemed unrealistic given the current staffing challenges and added more pressure on the already overburdened healthcare team.

Matters further deteriorated when executives from the corporate level made in-person hospital visits, during which they employed bullying tactics that served to intimidate and demoralize the staff. These aggressive approaches not only exacerbated the existing tensions but also contributed to an increasingly hostile and toxic work environment. On several occasions, the CMO witnessed distressing episodes of emotional and verbal bullying. In these instances, a corporate-level vice president openly belittled both executive and nursing staff, insinuating that these dedicated professionals were falling short of the standards set by the corporate hierarchy. These confrontations further poisoned the workplace atmosphere, deepening the rift between the corporate leadership and the hospital staff who were on the front lines of patient care. This workplace bullying served as a deflection from the urgent staffing and patient care problems that needed immediate attention. Alarmingly, it appeared that these aggressive tactics took precedence over clinical and ethical concerns, potentially constituting workplace violations. Such behavior from upper management further eroded trust and morale among staff members, making it even more challenging to address and resolve the pressing issues facing the hospital. Eventually, these persistent examples of dysfunctional leadership contributed significantly to escalating job stress among the staff. They obliterated any remaining sense that employees would be treated fairly and equitably. In the end, this toxic environment severely eroded the staff’s commitment to their work, with many questioning the sustainability of continuing to work under such demoralizing conditions. Thus, the CMO recognized that certain “course corrections” were critically necessary. He understood that the VBH medical leadership mission needed to pivot and concentrate its efforts on addressing the interconnected problems of how to best enhance the quality of patient care, boost staff job engagement and satisfaction, and reduce the intentions of staff turnover. To navigate these complexities, the CMO knew he needed to conduct a thorough assessment of the work dynamics, staff, and organizational characteristics that were contributing to both staff and organizational well-being and a positive work environment. Further, he was committed to developing a deeper understanding of how these relationships could be leveraged to cultivate more effective, compassionate, and ethical leadership within the hospital.

## 3. Methodology

To have a better understanding of the challenges faced in mental health settings, especially at VBH, and to observe the prominence of main workplace themes so that appropriate interventions focused on mindfulness and compassion could be developed [71], the CMO initiated a study that employs a phenomenological mixed method (PMM). PMM combines and integrates quantitative survey data and qualitative information in the form of narrative statements about the employees’ lived experiences [72]. Participants were reached out to through the American Psychiatric Nursing Association (APNA—Washington chapter) or via hospital staff rosters. Those who agreed to participate were contacted through electronic written communications. Participants were informed about the research, and their consent was obtained. Potential participants were required to have served as psychiatric nurses or mental health technicians during the COVID-19 pandemic and to have been in their position for at least 1 year. The recommendation to have a sample size for a qualitative study between 5 and 25 people was followed [73].

Utilizing his business background, the CMO worked directly with the HR department to design and disseminate a questionnaire. This tool was crafted to assess various work, staff, and organizational characteristics, aiming to pinpoint the elements contributing to both employee well-being and a positive workplace environment. The initial phase of curating this tool involved three key steps.

First, the CMO’s examination of exit interviews, sourced from the human resources department of VBH Inpatient Psychiatric Hospital, revealed concerning trends. He observed that past employees frequently cited decreased staffing levels, diminished job satisfaction, and a poor quality of life as significant issues. These factors were noted to have adverse effects on the quality of patient care. Moreover, the CMO identified increased interpersonal conflicts and a pervasive “lack of a caring culture” as primary reasons cited by critical staff members for their decision to depart from the organization.

Second, taking this information into account, and as part of the questionnaire design process, VBH human resources and the CMO conducted an informal pre-survey to gain insights into the reasons why employees were leaving their jobs. This pre-survey revealed that job dissatisfaction was a prevalent and recurring factor among the employees.

Third, synthesizing the information from both the exit interviews and the pre-survey results, the CMO drew inspiration from the approach employed by a well-established Canadian financial and insurance company that sought insights on employee retention and well-being. He developed a 5-point Likert scale questionnaire that was adapted from the Work Design Questionnaire (WDQ) [74]. This tool is designed to collect data related to several salient positive and negative employee- and work-related factors, including (a) task characteristics, (b) knowledge characteristics, (c) social characteristics, (d) justice aspects, (e) behavioral aspects, and (f) attitudinal variables. A detailed breakdown of the organizational, psychological, and attitudinal factors captured by the WDQ can be found in Table 1.

Out of the 21 staff members, 19 individuals provided responses to the initial questionnaire. This sample included a diverse range of roles within the hospital, including mental health technicians, nurses, mid-level and upper-level management, and psychiatric providers (MDs and Psychiatric Mental Health Nurse Practitioners PMHNPs).

## 4. Survey Results

The data collected through the questionnaire are analyzed using SAS 9.4 software (SAS Institute Inc., Cary, NC, USA). First, to gather insights into the characteristics of employees, we report demographic information (obtained from Questions Q1–Q7 of the questionnaire). We observed that the average age range of VBH staff members was between 30 and 45 years old, representing 44.4% of the respondents. Overall, 22.2% were between 20 and 30 years of age. Most respondents identified as female (83.3%). In terms of ethnicity, 50.0% described themselves as Caucasian, 22.2% reported mixed heritage, 11.1% identified as African American, 11.1% as Asian, and 5.6% as Latino or Hispanic.

Regarding educational qualifications, approximately one-third of respondents (36.4%) held a high school diploma and worked as mental health technicians, 27.2% had an Associate degree and worked in nursing, 50.0% had a college diploma and worked in nursing, and 18.8% had a Master’s, MD, or PhD equivalent. When it came to income, 27.8% earned between $30,000 and $49,999, 44.4% earned between $50,000 and $99,999, 5.6% earned between $100,000 and $124,999, and 22.2% earned between $150,000 and $199,999. On average, respondents had been in their current positions for 24–30 months (around 2 and a half years), with 63.2% reporting a tenure of one year or less. The mean working hours per week across all job positions was 45.21. The demographic information details are presented in Table 2.

As the next step, we categorize responses into two groups for comparison: Group 1 consists of employees who indicated that they do not intend to quit their current job for another job (*n* = 10), while Group 2 comprises employees who expressed their intention to quit for another job (*n* = 9). Our aim is to understand whether the responses of the employees that indicated that they intended to quit their job differed significantly from those that indicated that they did not intend to quit their job. We conducted the Wilcoxon rank sum test and Fisher’s exact test, generating *p*-values and table results. In Table 2, we present the results of the analysis pertaining to questions related to employee characteristics. The data indicate that there is no significant difference between the two groups along these dimensions.

Next, we conducted similar analyses for the remainder of the questions in the questionnaire (questions Q8–Q46). In the remainder of this section, we focus on the findings that indicate significant differences in the responses of the two groups.

Table 3 and Table 4 provide information on the statements that the employees who intended to leave their jobs are more likely to agree or disagree with compared to those who did not intend to leave their jobs.

## 5. Thematic Results

Considering the observations depicted in Section 5, we developed thematic insights into the relationships between organizational factors, job demands, burnout, affective commitment, and job satisfaction. We included discrete quotations to help clarify the respondents’ perspective [75].

### 5.1. Organizational Factors and Their Relationships to Job Demands

We observe from our small-scale study that a lack of autonomy in one’s job is associated with their inclination to leave it. Specifically, individuals who feel they do not possess adequate autonomy or the freedom to influence various aspects of their job—like scheduling, work methods, the nature of tasks, and overarching objectives—tend to express a stronger intent to resign. One staff member’s remark, “The work never ends!” came after strongly disagreeing with the notion that their job has a clear beginning and end and agreeing with feeling pressured by their employer. Another comment, “Utilization review controls everything,” was from an individual who disagreed with feeling pressured by their employer to meet objectives.

These comments underscore the importance of task identity, an intrinsic motivator that enables employees to take charge of their tasks from start to finish. This autonomy provides a sense of accomplishment and meaning in their work [76,77]. Task identity not only enhances job satisfaction but also imbues the task with a deeper significance.

Additionally, there is a discernible dissatisfaction sentiment regarding justice perceptions among those contemplating job resignation. Many vehemently disagreed with the presence of distributive justice, which signifies fair compensation for commendable work and rewards based on accomplishments. The sentiment of lacking procedural justice, the fair and unbiased application of processes, was also prominent. This sentiment was encapsulated in a poignant statement by one respondent: “Taking my money away from my weekend is my last straw”, starkly contrasting with those content in their positions.

These findings relate back to the organizational factors gauged by WDQ. These are markers for job demands, characterized as the “physical, social, and organizational aspects of a job that necessitate sustained physical or mental exertion” [78]. Overwhelming job demands often manifest as extended work hours, understaffing, or an absence of coworker support. Organizational reasons contributing to these heightened demands might encompass a deficit in administrative backing, prevailing injustices, and perceived unfairness. Employees in psychiatric hospitals confront these escalating demands with limited resources, inducing increased work stress, feelings of powerlessness, and perceptions of injustice. Achieving set objectives epitomizes work autonomy, and receiving fair treatment and compensation equates to feeling esteemed at work. The pervasive feelings of compensation inequity, inconsistent rule application, and devaluation suggest that the intense job demands at VBH might be precipitating these sentiments.

### 5.2. Burnout, Affective Commitment, and Their Relationship to Job Satisfaction

Questions of burnout have to do with a set of diverse symptoms including feeling stressed out, exhausted, and more negative or cynical. The differences between the employees that intended to leave their job and those that did not were most apparent in their agreement with the statements (i) “Some days I feel tired even before I get to work” and (ii) “During my work days I often feel emotionally drained (burnout variables) and their disagreement with the statements (i) “I am very satisfied with my current job” (job satisfaction), “Stress levels are manageable at work” (burnout), and “I am proud to belong to this organization” (affective commitment). There were two negative comments relating to manageable stress levels: “We are always understaffed” and “I get distracted when the workload increases”.

To visually depict the emotional state of the participants, we delved deeper into the anonymous responses given by study participants to an open-ended question seeking their feedback and comments. A word-cloud diagram derived from the aggregated responses that showcases psychological variables such as emotional and motivational terms is depicted in Figure 3.

Analyzing the twenty-one such terms identified, ten conveyed a negative outlook, while another ten suggested a positive perspective by the individuals. In descending order of frequency, the most frequently mentioned terms were: (1) frustration, (2) overwhelmed, (3) anxiety, (4) burnout. This sequence indicates a predominantly negative perception of the work environment. Extending our observation to the top ten words (excluding the neutral term “leadership”), the list consists of six negative terms with the inclusion of “disconnected” and “moral distress”. Conversely, the four positive terms are “empathy”, “compassion”, “satisfaction”, and “confidence”. We believe that these outcomes parallel our understanding of the drivers that impact participants’ view of various organizational, job, and attitudinal factors, gathered through the qualitative survey. In fact, through this analysis, we see that addressing the negative associations the participants had towards these factors is critical for improving staff retention and the effectiveness of delivery of care.

## 6. Discussion

The CMO, who had held a leadership position at a large psychiatric hospital during the first two waves of COVID-19, had already witnessed firsthand the devastating effects of staff stress and organizational dysfunction leading to job dissatisfaction, burnout, and resignations. When he assumed the role of CMO at VBH in 2021, he brought forward leadership expertise, solidifying his belief that an imbalance in “compassion equilibrium” had undermined numerous hospital work environments. Determined to confront this issue directly, the CMO commenced his tenure by initiating a study that aimed to comprehend how staff and workplace dynamics could potentially erode a positive and compassionate work environment. His goal was to introduce interventions that would enhance staff performance and well-being.

Qualitative and quantitative information gathered helped the CMO understand these dynamics and revealed that VBH psychiatric staff faced escalating job demands without adequate resources. This led to increased work pressure, feelings of being overwhelmed due to a perceived lack of control, and a prevailing sense of injustice. These challenges hindered the staff’s ability to meet both work and financial compensation objectives. A significant source of this perceived injustice was the belief in unequal financial compensation relative to the quantity or quality of work performed. Compounding these concerns was a perceived lack of procedural justice; staff believed that organizational rules were not applied fairly. Furthermore, diminished job satisfaction, stemming from feelings of being undervalued and experiencing burnout, eroded the psychiatric staff’s emotional commitment to VBH, fueling their intentions to depart from the organization.

In the backdrop of an already-strained psychiatric healthcare system, where state psychiatric hospitals were grappling with mounting financial losses, the cumulative stress for psychiatric staff reached a critical point. This led to a significant reduction in organizational resources and waning staff engagement [79]. Precisely when hospital organizations required heightened staff commitment, the workforce confronted an overwhelming feeling of burnout and emotional detachment.

It is important to qualify the meaning of burnout, as different people respond differently to job and personal stress, which may make them prone to burnout or more resilient. Recent studies have begun to explore the efficacy of specific psychological interventions to help the most vulnerable populations affected by the COVID-19 pandemic, including frontline nurses, adolescents, palliative care workers, college students, and university employees [12,80,81,82]. These interventions vary from psychological support [83] to mindfulness- and self-compassion-based techniques [84] to social connectedness [85] to prevention programs that target cognitive reappraisal strategy and resilience [24] and have shown promise improving burnout.

After a review of the feedback from the staff and management, the outcome of the analysis of the quantitative and qualitative data of this mixed methods study led to the development of a conceptual model (see Figure 2) of the proposed relationships in the mental healthcare settings among the critical factors related to job demands, stress, and burnout and the employees’ intention to leave. The proposed relationships are illustrated in Figure 4 and detailed in Propositions 1–5.

Proposition 1a: In mental healthcare settings, the employee’s procedural justice is negatively associated with the employee’s burnout.

Proposition 1b: In mental healthcare settings, the employee’s distributive justice is negatively associated with the employee’s burnout.

Proposition 1c: In mental healthcare settings, the employee’s ability to achieve objectives is negatively associated with the employee’s burnout.

Proposition 2a: In mental healthcare settings, the work autonomy experienced by the employee is positively associated with their job satisfaction.

Proposition 2b: In mental healthcare settings, the affective commitment experienced by the employee is positively associated with their job satisfaction.

Proposition 3: In mental healthcare settings, the employee’s burnout is negatively associated with their job satisfaction.

Proposition 4: In mental healthcare settings, the employee’s burnout is negatively associated with their intent to leave the organization.

Proposition 5: In mental healthcare settings, the employee’s job satisfaction is positively associated with their intent to leave the organization.

The current healthcare environment is one where (i) leaders in nursing (CNO, CEO) and medicine (CMO) face significant leadership and management challenges due to social, policy, and financial instability, whose effects are expected to linger well into the future; (ii) transformations among leadership roles, such as the transition of CNOs into CEOs, require increased fluidity between clinical and administrative roles [86]; (iii) there is a greater need for a foundational understanding of the business of healthcare [87]; and (iv) a better understanding of dynamic and adaptive features of health systems, such as telehealth technologies, is expected. As such, the evolving healthcare landscape will require a more comprehensive understanding of which work factors need to change to minimize the negative effects of burnout on individual and family well-being [88].

The insights derived from our qualitative case studies, narratives, and a small-scale survey, coupled with the proposed framework, set the stage for an advanced understanding of the impact of organizational factors and burnout on employee intentions to resign, especially within the high-pressure context of for-profit psychiatric hospitals. This study aims to delve deeper into the intricate relationship between job satisfaction, burnout, organizational justice, and how these factors influence employees’ decisions to leave [26,65]. It underscores how a perceived lack of autonomy, unrealistic job objectives, and insufficient distributive and procedural justice serve as primary predictors of burnout, creating an environment prone to high turnover rates [74,78]. Furthermore, the discussion on the vital role of leadership in alleviating these effects, by nurturing a culture of empathy, compassion, and effective communication, provides essential insights for healthcare administrators focused on enhancing staff retention and the quality of patient care [62,66]. In doing so, it addresses a significant gap in the literature by connecting organizational behavior theories with the practical challenges healthcare workers face in psychiatric settings amidst a global health crisis, thus paving the way for future research and interventions designed to strengthen the resilience of healthcare systems and their workforce.

## 7. Conclusions

The CMO recognized that it was critical for him to develop a better understanding of these proposed relationships and validate them through a larger-scale study. At the same time, faced with the leadership challenges detailed above, his immediate next step at this critical juncture was to engage both VBH staff and management to navigate this strenuous phase. With his foundation as an instructor of mindfulness practices and his academic inclination towards compassionate leadership in business, he endeavored to cultivate a more positive healthcare ambiance. He introduced a pilot self-compassion intervention aimed at mitigating burnout while enhancing staff communication and resilience. Collaborating with a compassion and mindfulness instructor, who was also a lawyer, mediator, and divinity scholar, they orchestrated a one-day hospital behavioral intervention grounded in the methodologies of Dr. Kristin Neff. Even though the participant count was limited (*n* = 9) and primarily comprised upper-level management, there were discernible improvements in mindfulness and perspective-taking. Despite the discussions being somewhat narrow in reach and not fully addressing VBH’s ongoing challenges, the CMO identified a promising pathway that could significantly benefit healthcare environments. Future work is aimed at focusing on examining these relationships in diverse mental healthcare landscapes, aiming to devise practical strategies that elevate employee well-being and satisfaction, which, in turn, should bolster employee retention in workplaces. In addition, as a benefit at the more strategic level, the planned outcome is to bring the relationships among the executive administration, CEO, CMO, and staff tasked with executing the operational activities to be structured closer to Sonnenberg’s proposed model (Figure 1). As the CMO understood, this could be achieved by disseminating the “compassionate caring” mindset at all levels of the organization.

This study has several inherent limitations. Factors such as diversity, religion and spirituality, coping style, and locus of control were not examined. Additionally, the influence of leadership style in alleviating challenges like interprofessional team miscommunication, burnout, job stress, and work dissatisfaction were not systematically investigated. To both validate our current psychiatric hospital workforce model and formulate a more encompassing and efficacious organizational leadership approach, it is imperative to investigate these additional dimensions. Yet, emerging evidence suggests that beyond mere knowledge application, a compassionate understanding is pivotal in fostering positive collaborative dynamics in healthcare environments [89,90]. In subsequent studies, the focus should pivot to determining how compassionate leadership can nurture staff well-being and bolster employee support in the for-profit psychiatric hospital domain.

## Figures and Tables

**Figure 1 ijerph-21-00484-f001:**
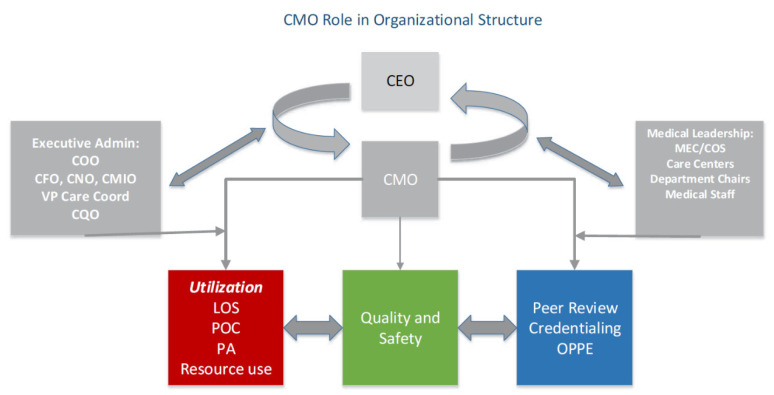
Sonnenberg’s model of the responsibilities of CMOs and the staff in hospital settings [70].

**Figure 2 ijerph-21-00484-f002:**
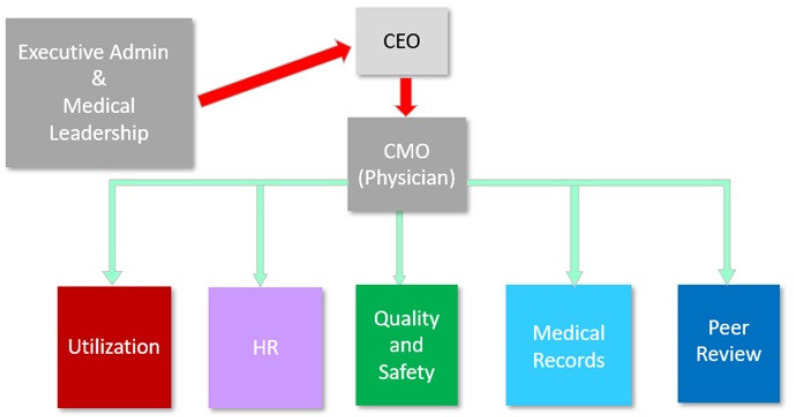
The responsibilities of the CMO and the staff at VBH.

**Figure 3 ijerph-21-00484-f003:**
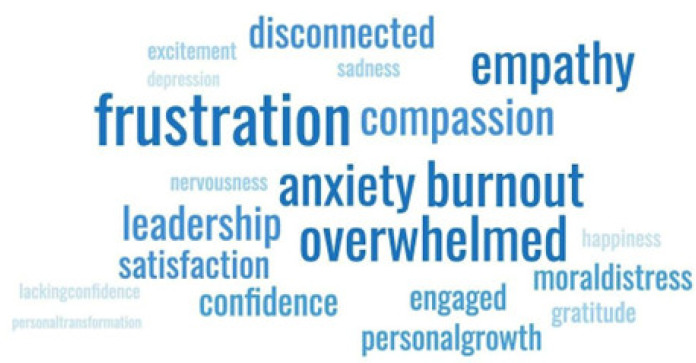
Word-cloud diagram summarizing the emotional state of the study participants.

**Figure 4 ijerph-21-00484-f004:**
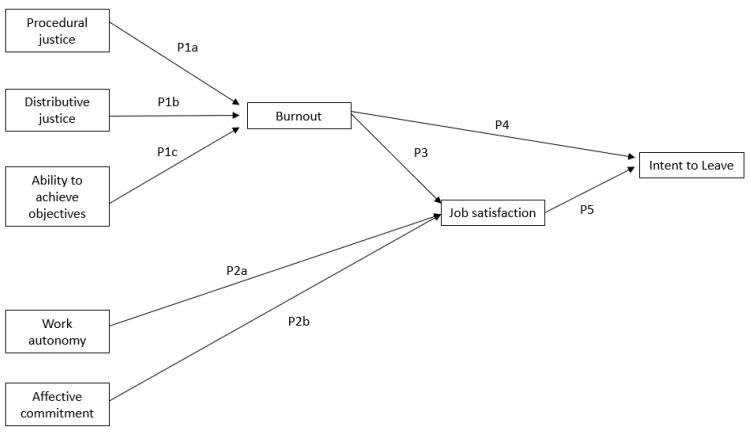
Proposed relationships among organizational, psychological, and attitudinal factors and the intent to leave.

**Table 1 ijerph-21-00484-t001:** Organizational, psychological, and attitudinal factors captured by WDQ.

Organizational Factors			
Job related factors: - Work autonomy - Task variety - Task significance - Task identity - Ability to achieve objectives	Knowledge Characteristics: - Task Characteristics - Information Processing - Skill Variety	Justice Aspects: - Distributive justice - Procedural justice - Informational justice - Relational justice	Social Characteristics: - Social support - Client feedback - Manager feedback
Psychological Factors			
Burnout			
Attitudinal Factors			
- Overall Job Satisfaction	- Affective Commitment		- Quit Intentions

**Table 2 ijerph-21-00484-t002:** Descriptive statistics of employee characteristics.

Variables	I Do not Intend to Quit My Job for Another Job (*n* = 10)	I Intend to Quit My Job for Another Job (*n* = 9)	Total (*n* = 19)	*p*-Value
**(1)** **Age**				
N Missing	1	0	1	0.62
20–30 yr	3 (33.3%)	1 (11.1%)	4 (22.2%)	
30–45 yr	3 (33.3%)	5 (55.6%)	8 (44.4%)	
45–60 yr	3 (33.3%)	3 (33.3%)	6 (33.3%)	
**(2)** **Gender**				
N Missing	1	0	1	0.99
Female	8 (88.9%)	7 (77.8%)	15 (83.3%)	
Male	1 (11.1%)	2 (22.2%)	3 (16.7%)	
**(3)** **Ethnicity**				
N Missing	1	0	1	0.25
Asian	2 (22.2%)	0	2 (11.1%)	
Black or African American	0	2 (22.2%)	2 (11.1%)	
Hispanic or Latin	1 (11.1%)	0	1 (5.6%)	
Mixed	1 (11.1%)	3 (33.3%)	4 (22.2%)	
White	5 (55.6%)	4 (44.4%)	9 (50.0%)	
**(4)** **Annual Pay**				
N Missing	1	0	1	0.14
1. 30,000–49,999	1 (11.1%)	4 (44.4%)	5 (27.8%)	
2. 50,000–99,999	6 (66.7%)	2 (22.2%)	8 (44.4%)	
3. 100,000–124,999	1 (11.1%)	0	1 (5.6%)	
4. 150,000–199,999	1 (11.1%)	3 (33.3%)	4 (22.2%)	
**(5)** **Length of Current Position**				
1. <1 Year	8 (80.0%)	4 (44.4%)	12 (63.2%)	0.34
2. 1–2 Year	1 (10.0%)	3 (33.3%)	4 (21.1%)	
3. >2 Years	1 (10.0%)	2 (22.2%)	3 (15.8%)	
**(6)** **Position**				
N Missing	4	4	8	0.14
Entry Level	1 (16.7%)	0	1 (9.1%)	
Mid Level	0	1 (20.0%)	1 (9.1%)	
Nursing	3 (50.0%)	0	3 (27.3%)	
Physician	0	2 (40.0%)	2 (18.2%)	
Technician	2 (33.3%)	2 (40.0%)	4 (36.4%)	
**(7)** **Working hours per week**				
Mean (SD)	44.4 (5.0)	45.9 (7.2)	45.2 (6.1)	0.90
Median (Q1, Q3)	45.0 (40.0, 48.0)	44.5 (42.0, 49.0)	45.0 (40.0, 48.0)	

**Table 3 ijerph-21-00484-t003:** Statements that employees who intended to leave their jobs are more likely to agree with compared to those who did not intend to leave their jobs.

Question Number	Question Details	Variable	*p*-Value
Q18	I feel pressured by my employer to meet my objectives	Ability to Achieve Objectives	0.05 **
Q42	Some days I feel tired even before I get to work	Burnout	0.01 **
Q43	During my workdays I often feel emotionally drained	Burnout	0.03 **

** *p* ≤ 0.05.

**Table 4 ijerph-21-00484-t004:** Statements that employees who intended to leave their jobs are more likely to disagree with compared to those who did not intend to leave their jobs.

Question Number	Question Details	Variable	*p*-Value
Q 11	I have some control over what I am supposed to accomplish	Work Autonomy	0.023 **
Q16	My job allows me to complete work I start	Ability to Achieve Objectives	0.002 ***
Q19	I feel that my objectives are realistic	Ability to Achieve Objectives	0.04 **
Q20	Reaching my objective gives me access to compensation (salary and bonuses)	Distributive Justice	<0.0001 ***
Q35	I am fairly compensated for work well done	Distributive Justice	0.02 **
Q37	The procedures are applied consistently and uniformly	Procedural Justice	0.05 *
Q45	I am very satisfied with my current job	Job Satisfaction	0.01 **
Q46	Stress levels are manageable at work	Burnout	0.04 **
Q48	I am proud to belong to this organization	Affective Commitment	0.003 ***

* *p* ≤ 0.10; ** *p* ≤ 0.05; *** *p* < 0.01.

## Data Availability

Data are contained within the article.

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
