# Peer review of "Insights into the Impact of Organizational Factors and Burnout on the Employees of a For-Profit Psychiatric Hospital during the Third Wave of the COVID-19 Pandemic"

_ijerph, 2024, doi:10.3390/ijerph21040484_

Round 1
Reviewer 1 Report
Comments and Suggestions for Authors
The paper provides insightful findings about the underlying organizational, job, and attitudinal factors that drove employees' intentions to resign during the third wave of the COVID-19 pandemic at a mental health hospital. Anyway, something still needs improvement. I encourage the authors to consider the comments given below and revise the paper accordingly in order to enhance the overall quality and completeness of the paper.
(1) Since research on the psychological impacts of the COVID-19 has been quite saturated, it is very important for the authors to clearly state in the introduction about the research gaps and the key contributions that their finding provides to fill the gaps. The uniqueness of the paper must be emphasized specifically in the introduction.
(2) The literature review needs to include the papers that provide the integrative model about the personal characteristics and work characteristics that need to be considered together to give a realistic explanation of why employees experienced stress during the pandemic. Such integrative framework can be found in the papers listed below, which need to be integrated as the additional references in the literature review section.
a. How Does Mindfulness Help University Employees Cope with Emotional Exhaustion during the COVID-19 Crisis? The Mediating Role of Psychological Hardiness and the Moderating Effect of Workload, Scandinavian Journal of Psychology. 63(5), 449-461. https://doi.org/10.1111/sjop.12826
b. Stressors of COVID-19 and stress consequences: The mediating role of rumination and the moderating role of psychological support. Children and Youth Services Review, 118, 105466. https://doi.org/10.1016/j.childyouth.2020.105466
(3) Given that different people may not be prone to burnout to the same extent, the authors need to discuss in some more detail how individual differences could affect the outcomes. In fact, the authors already mentioned a little bit about this issue in the limitation section. However, some more explanation is required how individual differences could matter.
(4) The knowledge contributions of the study need to be emphasized in the discussion section.
Reviewer 2 Report
Comments and Suggestions for Authors
The presented article is extremely interesting and provides a different perspective on how the COVID pandemic affected various healthcare resources, in this case, psychiatric centers.
Below, we present a series of aspects that would help improve the presented work:
It would be relevant to incorporate some data about the number of participants in the study in the abstract. Neither is it pointed out how the data analysis has been conducted.
The introduction, describing the study, is unnecessary. The introduction should place the reader in what is known about the study topic.
In the first part of point 2 (CMO Assessment: COVID-19), the authors make statements that cannot be verified and are more like personal assessments without offering data to corroborate those statements. For example: "During most of the year, there was a revolving door of Chief Nursing Officers (CNO), Chief Executive Officers (CEO), and Chief Medical Officers (CMO), which only served to increase organizational chaos and weaken staff morale. Subsequently, during 2020, the world experienced the onset and escalation of the COVID-19 pandemic."
There are many similar assertions in the same vein that make the text lose rigor and scientific sense.
Methodologically, certain statements cannot be made (let alone certain types of statistical analysis) with such a small sample size.
Round 2
Reviewer 1 Report
Comments and Suggestions for Authors
The authors did a satisfactory revision. The quality of the paper is now adequately improved. There is no further comment.
Author Response
We sincerely thank the reviewers for their additional editorial comments on our manuscript. Specifically, we have carefully addressed the issue of using only one place after the decimal. We have made these editorial changes to all tables and texts where appropriate.
We trust these changes are satisfactory to the reviewers and editorial board members.
Reviewer 2 Report
Comments and Suggestions for Authors
The article "Insights into the Impact of Organizational Factors and Burnout on the Employees of a For-Profit Psychiatric Hospital during the Third Wave of the COVID-19 Pandemic" attempts to provide insight into the challenges faced by mental health workers in a psychiatric hospital during the intensification of the COVID-19 pandemic. In the study, which follows a mixed methodology, incorporating both qualitative and quantitative data, it is notable for its attempt to disentangle the interrelationship between organizational factors, burnout, job satisfaction, and employee resignation intention.
Nevertheless, the study again continues to present important limitations. The small sample size (n=19) may limit the generalizability of the results to other contexts or institutions. There are still important flaws in the data and in the qualitative analysis of the data.
Author Response

(The authors gave the same response as above.)
